# Computational Identification and 3D Morphological Characterization of Renal Glomeruli in Optically Cleared Murine Kidneys

**DOI:** 10.3390/s21227440

**Published:** 2021-11-09

**Authors:** Nabil Nicolas, Nour Nicolas, Etienne Roux

**Affiliations:** INSERM, Biologie Des Maladies Cardiovasculaires, University Bordeaux, U1034, F-33600 Pessac, France; nour.nicolas@etu.u-bordeaux.fr (N.N.); etienne.roux@u-bordeaux.fr (E.R.)

**Keywords:** renal glomeruli, optical clearing, light sheet microscopy, 3D imaging, 3D image processing, 3D morphological characterization

## Abstract

The aim of this study was to establish an accessible methodology for the objective identification and 3D morphological characterization of renal glomeruli in mice. 3D imaging of the renal cortex was performed by light sheet microscopy on iDISCO+ optical cleared kidneys of six C57BL/6J mice after labelling of the capillary endothelium by lectin injection. 3D images were processed with the open source software ImageJ, and statistical analysis done with GraphPad Prism. Non-visual delimitation of the external surface of the glomeruli was ensured by greyscale-based thresholding, the value of which was determined from the statistical analysis of the voxel frequency distribution. Exclusion of false-positive identification was done by successive volume- and shape-based segmentation. Renal glomeruli were characterized by their number, surface area, volume, and compactness. Average data were expressed as mean ± SD. The number of glomeruli was equal to 283 ± 35 per mm^3^ of renal tissue, representing 1.78 ± 0.49% of the tissue volume. The surface area, volume and compactness were equal to 20,830 ± 6200 µm², 62,280 ± 14,000 µm^3^ and 0.068 ± 0.026, respectively. The proposed standardized methodology allows the identification of the renal glomeruli and their 3D morphological characterization, and is easily accessible for biologists.

## 1. Introduction

Kidneys are vital richly vascularized organs with a main function consisting of the ultrafiltration of the blood. This results in the production of the primitive form of the urine, allowing in fine the regulation of the organism’s water, ions, and metabolite levels [1]. This main function is performed by the renal cortex glomeruli, that are tufts of capillaries constituted with a fenestrated endothelium allowing the blood ultrafiltration. Alteration of the glomerulus capillary network can impact the glomerular blood flow and thus the ultrafiltration, resulting in glomerular dysfunction [2,3]. Glomerular dysfunction can not only be responsible for kidney failure, but also cardiovascular diseases [4].

Identification of the renal glomeruli structure is hence of critical importance for the understanding of physiological renal glomeruli functional properties and the physiopathology of several renal and cardiovascular dysfunctions. Historically, most of the undergone studies that intended to analyze human and animal renal glomeruli have not determined the specific characterization of their 3D structure. In humans, in vivo structural study of the renal glomeruli is impossible as echography, the main technique used for diagnosis, is not resolutive enough [5]. Human renal cortex biopsies have been done on living kidney donors and cadaver donors to study the glomerular structure via 2D, not 3D, electron microscopy and light microscopy [6,7]. In rodents, a technique has been established for the extraction of intact glomeruli from renal tissue, but was not associated with techniques allowing the 3D structural characterization of the isolated glomeruli [3,8]. 3D glomerular structure has been reconstructed from glomerulus serial 2D sectional images taken by electron microscopy and photonic microscopy [9,10]. Recently, MRI has been performed on mice to count the glomeruli and calculate their volume from 3D images [11]. While interesting, these different studies did not provide quantitative characterization of the complex surface of the glomeruli. 

A critical issue of the imaging techniques is the objective and automated identification of the boundaries between the objects of interest, i.e., the renal glomeruli, and their surrounding environment. Indeed, in the raw images obtained from the imaging techniques, the glomeruli are not separated from the tissue in which they are embedded. Contrast-based segmentation procedures are needed to identify the glomerulus-background interface. As a consequence, the identification of the surface of the objects of interest is critically dependent on the segmentation process, and may lead to substantial differences depending on the techniques used. Approximative determination of the object–background interface is not an issue when the objective of the study is to identify the core of the objects of interest, but becomes a source of potential bias and mismeasurement when the aim is to characterize the surface of the objects and their 3D morphology. The aim of our study was to establish an open-access standardized methodology for operator-independent identification of the glomerulus–background interface and quantitative 3D morphological characterization of renal cortical glomeruli in 3D imaged mouse kidneys.

For 3D imaging, the strategy was to use 3D photonic imaging to visualize the renal glomeruli on intact mice kidneys. A major problem is light scattering that limits light penetration in the tissue. Clearing methods that homogenize the refractive index have been developed in the past decade to overcome this difficulty [12]. Among the several techniques available for optical clearing, we have chosen the iDISCO+ method [13]. Its main advantages are easy access, short clearing time and high clearing capacity. Labelling of the capillary network was ensured by lectin coupled to a fluorophore, as lectin has been shown to be a specific marker of the capillary endothelium [14]. Cleared samples were imaged by light sheet microscopy (LSM) that allows large 3D visualization in intact samples [12].

For 3D image processing, the methodology used to objectivize the glomerulus-background interface was based on the statistical analysis of the greyscale frequency distribution of the voxel population, which allowed the determination of a greyscale threshold value independent from any visual estimate. The subsequent greyscale segmentation ensured the extraction of the lectin-labelled objects, most of them glomeruli, from the background tissue. This step was followed by two other successive segmentations based on the volume and the shape of the objects, respectively, in order to exclude lectin-labelled but non-glomerular objects and retain the glomeruli. Their number was calculated and their 3D morphology was characterized by their individual and average surface area, volume and compactness.

## 2. Materials and Methods

### 2.1. Animal Model

Experiments were performed on 8-week-old C57BL/6J mice (*n* = 6). Food and water were available ad libitum, with a 12-h dark/light cycle. Each mouse received a retro-orbital intravenous 100 µL injection of lectin coupled to a fluorophore (*Lycopersicon esculentum* Lectin DyLight649^®^ 1 mg/mL, Vector Labs) with an insulin syringe. Ten minutes after the injection, an intraperitoneal 100 µL injection of isosorbide dinitrate (Risordan^®^ 10 mg/10 mL, Medisol) was realized with a 25 G needle to dilate the vessels. The mouse was then euthanized by an intraperitoneal injection of 300 µL sodium pentobarbital (Exagon^®^ à 400 mg/mL, Axience) diluted in physiological saline solution. After death, a sternotomy was realized to catheterize the left ventricle. A perfusion of physiological solution at 80 mmHg pressure during 3 min, was done to remove the blood from the vasculature. A second perfusion of 4% formalin (10% neutral buffered formalin, DiaPath), was done to fix the tissues. The left kidney was delicately removed and placed in paraformalin overnight at 4 °C. A negative control was done is similar conditions, but without lectin injection.

### 2.2. Optical Clearing

Optical clearing of the tissues was performed using the iDISCO+ method [13]. The technique consists of a methanol pre-treatment, followed by methanol and dichloromethane permeabilization and lipid removal, and RI matching in dibenzyl ether.

Each kidney was pre-treated by successive immersions in 20%, 40%, 60%, and 80% methanol solutions for 1 h each at room temperature, then left overnight in a pure methanol solution (Methanol ≥ 99.5%, GPR RECTAPUR^®^, VWR Chemicals, Fontenay-sous-Bois, France) at room temperature. Permeabilization and lipid removal were then performed in a 2/3 dichloromethane (Dichloromethane, anhydro ≥ 99.8%, with 40–150 ppm amylene, Sigma-Aldrich, Lyon, France) and 1/3 methanol solution for 3 h at room temperature. The remaining methanol was removed by two 15-min baths in pure dichloromethane solutions at room temperature. Last, RI matching was done by leaving the sample in a dibenzyl ether solution (Benzyl ether 98%, Sigma-Aldrich) for a few hours at room temperature, then kept at 4 °C until imaging. 

The iDISCO+ technique has been shown to generate a shrinkage of the tissues [12]. On each sample, the shrinkage was estimated from two orthogonal 2D-images taken before and after clearing. Length (*l*), width (*w*), and thickness (*t*) were measured using ImageJ/Fiji software. These measurements were used to calculate an estimated kidney volume, considered as an ellipsoid, thanks to the following equation:(1)$$\mathrm{Kidney}\;\mathrm{Volume}=\frac{4}{3}\times \pi \times \frac{l}{2}\times \frac{w}{2}\times \frac{t}{2}$$

The shrinkage percentage was calculated for the length, the width, the thickness and the ellipsoidal volume as the ratio of each parameter value after and before optical clearing on the parameter value before optical clearing.

### 2.3. Image Acquisition

Image acquisition was done by light sheet microscopy. Each kidney was mounted in ethyl cinnamate solution in a specific device adapted to the light sheet microscope. The ultramicroscopy was done using the system from LaVision BioTec (Bielefeld, Germany) equipped with a 640 nm (70 mW) laser line, a sCMOS Andor camera, and a 0.5 NA 2 × objective with a deeping lens, with 6.3 zoom. The tissue was illuminated laterally by three horizontal sheets. Laser power was set at 5%. Exposition time was 200 ms. Emitted fluorescence was collected at 690 nm. At the end of the acquisition, the stack was automatically reconstructed in 16-bit format. Renal cortex imaging was made by sampling a 1080 × 1280 × 300 µm parallelipipedic section. The system spatial resolution was 1 µm (x, y) and 4 µm (z). Step size used was 2 µm. Voxel dimensions were 0.5 µm (x, y) and 2 µm (z).

### 2.4. Image Processing

Commands and functions for each step of the image processing are given as Appendix A.

The stacks were converted to 8-bit format (256 grey levels) and image processing was performed using the open source ImageJ/Fiji software (ImageJ 2.1.0/1.53c/Java 1.8.0_66 (64-bit)) and 3D Suites plugin [15]. Data extraction was performed using the *3D Suites* plugin, which allowed the automatic identification of the objects and the calculation of morphological parameters.

#### 2.4.1. Grey-Level Segmentations and Glomerulus-Background Interface Determination

The principle of 3D ultramicroscopy on fluorescent-labelled structures is that the voxels corresponding to the labelled objects have an average greyscale value higher than the non-labelled structures, allowing greyscale threshold-based segmentation of the objects of interest. In our experiment, this was ensured by the specific labelling of the glomerular capillary meshwork by fluorophore-coupled lectin. Additionally, the non-labelled renal tissue exhibited a non-specific spontaneous moderate fluorescence, while the mounting solution had no self-fluorescence. As a result, the 3D images obtained after greyscale conversion were composed of three populations of voxels, corresponding, from lowest to higher greyscale values, to the mounting solution, the self-fluorescent tissue, and the lectin-labelled structures. A representative frequency distribution of the voxel greyscale values obtained from one kidney is shown in Figure 1a. This distribution shows the existence of the three above-mentioned populations corresponding, from 0 to 255 grey values, to (1) the image background, (2) the non-labelled renal tissue, and (3) the lectin-labelled glomerular capillary network, respectively. While the first and second voxel populations could be distinguished by the existence of an inflexion point in the curve, the second and third populations were overlapping. The continuum between the two voxel populations corresponding to the renal tissue and the lectin-labelled capillary network made it difficult to identify an objective greyscale threshold value to distinguish the highest greyscale voxels of the non-labelled tissue from the lowest greyscale voxels of the lectin-labelled glomerular capillaries. This is a critical issue, since this threshold determines the interface between the glomeruli and the surrounding tissue. The methodology used to fix this problem was based on the fact that the non-labelled tissue corresponded to a homogenous population of grey-scale voxels following a Gaussian normal distribution, as evidenced by control experiments done on non lectin-labelled kidneys (Figure 1c,d). Due to the mathematical properties of the normal distribution, the mean ± 3.29 SD interval includes 99% of the voxel population corresponding to the non-labelled renal tissue. Hence, on each sample, the greyscale frequency distribution curve of the voxels was generated. The first threshold, segmenting the renal tissue from the mounting device background, was set at the inflexion point of the distribution curve between the first and second population. For the segmentation of the lectin-labelled structure from the tissular background, the bell-shaped population corresponding to the self-fluorescent tissue was selected and fitted by the following Gaussian equation:(2)$$F=F_{max}\times e^{(-0.5\times \left(\left(X-mean\right)/SD{)}^{2}\right)}$$
where F is the frequency of the voxel greyscale value, $F_{max}$ is the maximal frequency value, X is the vowel greyscale value, *mean* is the mean value of the Gaussian distribution, and SD its standard deviation. A representative Gaussian fit is given in (Figure 1b). This allowed determination of the mean and SD of the voxel population corresponding to the non-labelled renal tissue, and the threshold value was set at mean + 3.29 SD. Such a threshold value ensured the exclusion of 99.5% of the voxels that did not correspond to the lectin-labelled structures.

#### 2.4.2. Filtering

After binarization of the resulting segmented images, a 3D median filter set at three voxels for (x, y, z) was applied to suppress artefactual isolated voxels.

#### 2.4.3. Removal of Lectin-Labelled Non-Glomerular Objects

The renal vasculature is characterized by the existence of two successive capillary meshworks, the order of which is, following the blood flow, (1) the glomerular capillaries, and (2) the peritubular ones. Following IV injection, lectin labels the glomerular capillaries first.

Additionally, due to its small size, it is partially ultrafiltrated, resulting in a drop of lectin concentration in the post-glomerular vasculature. However, part of the peritubular capillaries may be labelled. In addition, lectin may also label some arteriolar segments. As a consequence, grey-level segmentation resulted in the identification of lectin-labelled objects including the glomeruli and several non-glomerular objects. These non-glomerular structures were distinguishable by their volume or their shape, and these morphological properties were used to remove these non-glomerular objects by two successive volume- and shape-based segmentations, respectively.

First, for each sample, after greyscale threshold-based segmentation, the volume of the lectin-labelled objects was determined using the *3D Suites* plugin, and the frequency distribution curve of the volumes was generated using GraphPad Prism^®^ software (Prism 9.0.1 version). A representative frequency distribution curve is given in Figure 1e The curve showed the existence of two main populations corresponding to a majority of (1) small lectin-labelled objects composed of non-glomerular objects, and (2) a minority of big lectin-labelled objects composed of non-glomerular and glomerular objects, separated by an inflexion point. The volume-based threshold was set at the inflexion point of the distribution curve between the first and second population, and a segmentation was performed to remove the small non-glomerular objects.

Volume segmentation resulted in images including glomeruli and non-glomerular objects of a similar volume range, but distinguishable by their shape. Indeed, glomeruli appeared more ellipsoidal than the non-glomerular objects, i.e., parts of peritubular meshwork and arteriolar segments. Hence, for each object of each sample, its real volume (Vobj) was calculated, as well as its ellipsoid volume (Vell), i.e., the volume of the smallest ellipsoid including the object, and their ratio (Vobj/Vell), using the *3D Suites* plugin. For each sample, the frequency distribution of the Vobj/Vell ratio was generated. A representative curve is given in Figure 1f. The distribution curve showed the existence of two main populations corresponding to (1) a minority of non-ellipsoid shaped objects corresponding to the non-glomerular objects, and (2) a majority of ellipsoid-shaped objects corresponding to the renal glomeruli, separated by an inflexion point. The threshold segmenting the non-ellipsoidal objects from the ellipsoidal ones was set at the inflexion point of the distribution curve between the first and second population.

#### 2.4.4. Segmentation Efficiency for the Individual Identification of the Glomeruli

For each sample, the three successive greyscale, volume and shape-based segmentations resulted in an image including the glomerular objects, each individually identified and defined by its external surface, and excluding non-glomerular lectin-labelled objects. In order to evaluate the ability of the method to perform a non erroneous identification of the glomeruli, control experiments were performed on the six samples to determine the percentage of false negative and false positive results of the method compared to the visual identification of the glomeruli. For this, in each sample, the glomeruli were visually identified, and the results of the visual identification compared with those of the segmentation-based method. Each object identified as a glomerulus by the segmentation process but not visually identified was considered as a false-positive result, and each glomerulus visually identified but not identified by the segmentation process was considered as a false-negative result. The number of false positive and false negative results was normalized to the number of visually identified glomeruli and expressed as a percentage.

### 2.5. Data Extraction and Calculated Parameters 

For each sample, the following parameters were calculated using the *3D Suites* plugin automatic measurement.

#### 2.5.1. Glomerular Volume Density

The total volume of renal tissue of each sample and the global glomerular volume were obtained from their respective segmented binary images. Each volume, in mm^3^, was calculated by multiplying the number of voxels corresponding to the objects of interest by the volume of one voxel in mm^3^. The glomerular volume density, expressed as %, was calculated according to the following equation:(3)glomerular volume density=glomerular volume/renal tissue volume

#### 2.5.2. Glomerular Numerical Density

The glomerular numerical density was defined as the ratio of the number of glomeruli obtained from the data extraction on the total renal tissue volume (in mm^3^) of the sample.

#### 2.5.3. Surface Area

The surface area of each glomerulus was obtained in µm² from the data extraction, and its mean value was calculated for each sample.

#### 2.5.4. Volume

The volume of each glomerulus was obtained in µm^3^ from the data extraction and its mean value was calculated for each sample.

#### 2.5.5. Compactness

The isoperimetric ratio is a dimensionless parameter defined by the surface^3^/volume² ratio of the 3D object of interest. It is equal to 36π for a perfect sphere [16]. The compactness, i.e., the isoperimetric ratio of the object normalized to the perfect sphere isoperimetric ratio, is defined as follows:(4)$$compactness=36\pi \times volume^{2}/surface^{3}$$

Its value varies from 0 to 1 for a perfect sphere. The compactness of each glomerulus was obtained from the data extraction, and its mean value was calculated for each sample.

### 2.6. Statistical Analysis

Statistical analyses were done using GraphPad Prism^®^ software (Prism 9.0.1 version). Shrinkage, percentage of false-positive and negative identifications, glomerular volume and numerical densities, average surface area, average volume and average compactness were expressed as mean ± standard deviation (SD). Shrinkage values in length, width and thickness were compared by the Kruskal–Wallis non-parametric test. The frequency distributions of the individual surface area, volume and compactness were determined and fitted by the following Gaussian equation:(5)$$F=A\times e^{\left(-0.5\times {\left(\left(P-µ\right)/\sigma \right)}^{2}\right)}$$
where P is the studied parameter, *F* is the relative frequency (in %), *A* is the maximal frequency, *µ* is the parameter mean value, and *σ* is the standard deviation.

## 3. Results

### 3.1. Shrinkage

Shrinkage values in length, width and thickness were 11.3 ± 4.6%, 10.2 ± 6.7% and 13.9 ± 12%, respectively. Statistical comparison showed no significant difference between these values, indicating that the shrinkage was isotropic. The ellipsoid volume shrinkage was 31.2 ± 13%.

### 3.2. Identification of the Objects of Interest and Segmentation Efficiency

The first greyscale threshold segmentation allowed the exclusion of the non-fluorescent background voxels, retaining the total renal tissue, including the self-fluorescent structures, the lectin-labelled objects, and punctual imaging artifacts. A representative grey-level segmented image of the renal tissue is shown in Figure 2a. The second greyscale threshold segmentation allowed the exclusion of the self-fluorescent voxels corresponding to the non lectin-labelled structure, and retaining the glomeruli and other lectin-labelled objects, and punctual artifacts (Figure 2b). Subsequent filtering, volume-based and shape-based segmentations resulted in the exclusion of around 90% of artifactual and lectin-labelled objects, supposedly non-glomerular structures such as arteriolar segments and peritubular capillaries. Representative images following volume-based and shape-based segmentations are given in Figure 2c,d, respectively. On each sample, the automated object identification following the segmentation process was compared with the visual identification of the glomeruli, identifiable by their brightness and “glomerular” shape. The percentage of false positive and false negative identifications was 3.83 ± 9.4% and 6.33 ± 8.6%, respectively.

### 3.3. Glomerular External Surface

The external surface of each glomerulus corresponds to its boundaries determined by the greyscale-based segmentation. Representative images of glomeruli from the six samples are shown in Figure 3. The images showed that glomeruli exhibited a globular morphology but with an irregular external surface.

### 3.4. Glomerular Volume and Numerical Density

Glomerular volume density was 1.78 ± 0.49% of the total tissue volume and the glomerular numerical density was 283 ± 35 glomeruli by mm^3^ of total renal tissue.

### 3.5. Surface Area, Volume and Compactness

Mean glomerular surface area, volume and compactness are given in Figure 4a,c,e, respectively. Additionally, since these parameters were identified for each glomerulus, the population of glomeruli was characterized by the frequency distribution curve of each parameter, and fitted by a Gaussian equation. Fitted frequency distribution curves for the surface area, the volume and the compactness are presented in Figure 4b,d,f respectively. The R^2^ values and estimated parameters of the fitting curves are given in Table 1.

## 4. Discussion

Our study has established a standardized method for the identification of the renal glomeruli and their 3D morphological characterization applicable to 3D photonic imaging of the lectin-labelled capillary network on optically-cleared kidneys from mice, a species largely used as models by biologists. This method generates both global and individual data that would allow 3D quantitative statistical analysis and comparison between samples.

### 4.1. 3D Imaging

The iDISCO+ optical clearing method used in this study is known, according to its developers, to induce an 11% volume shrinkage with minor deformation of the brain [13]. According to a previous study on heart, we established that, as in the brain, the volume shrinkage was minor, around 19%, without deformation [17]. Our results showed an estimated kidney volume shrinkage around 30%, without significant difference in each dimension, suggesting that, compared to the brain and heart, the iDISCO+ method induced a higher shrinkage, though isotropic, in kidneys. This might be explained by the water-rich composition of the kidney that is removed during the dehydration phase of the iDISCO+ protocol.

### 4.2. 3D Image Processing

Morphological characterization of 3D objects such as renal glomeruli requires (1) a correct identification of the objects of interest, and (2) an unbiased delimitation of their boundaries, i.e., their external surface that delineates the object from its surrounding environment. In our study, as in other techniques based on specific labelling, identification of the objects and delimitation of their boundaries are both grounded on the greyscale contrast between the object and the background. Due to the high average greyscale difference between the glomeruli and the renal tissue, visual discrimination ensures a correct identification of the glomeruli. Additionally, their specific shape, from which their name derives, allows the visual distinction between the glomeruli and non-glomerular labelled structures. Hence, visual identification of the glomeruli is basically unequivocal and remains a reference method, though highly time-consuming for large or numerous samples.

However, correct identification of the objects of interest does not ensure a correct delimitation on their boundary. Indeed, though the difference in average greyscale value between the glomeruli and the renal tissue is large, the peripheral voxels of the lectin-labelled objects have intermediate values, as shown by the continuum of voxel greyscale values in the frequency distribution curve. As a consequence, objective delimitation of the object boundary is difficult. Several filtering algorithms are available to enhance the contrast between high and low grey-level zones, but the choice of the adequate filter is usually based on a visual estimate of their results and, due to the poor greyscale resolution power of the eye, highly subjective. To overcome this difficulty and ensure a visually-independent delimitation of the object external surface, the method applied in this study is based on the statistical characterization of the non-lectin-labelled voxels, which constitute a homogenous population following a Gaussian distribution. The mathematical property of such a distribution allows the objective definition of a grey-scale threshold ensuring the exclusion of 99.5% of the voxels corresponding to the non-labelled structure. Hence, though the voxel distribution may vary from sample to sample, and despite the overlap of voxel populations of non-labelled and lectin-labelled structures, this method ensured that, for each sample, the same proportion of non-capillary renal tissue was excluded from the segmented image, and, hence, an objective delimitation of the boundaries of the lectin-labelled objects.

The greyscale-based segmentation determined (1) the delimitation of the external surface of the lectin-labelled objects, and (2) their identification; however, among them, there were a large number of non-glomerular objects, i.e., false-positive identifications. At that stage of the segmentation process, there were indeed 10,000 objects identified on average in each sample, including for 99% lectin-labelled non-glomerular objects and for around 1% of the lectin-labelled renal glomeruli. Almost all of the lectin-labelled non-glomerular objects were characterized by a very small volume, and could hence be excluded by a volume-based segmentation, leading to around one hundred objects left per sample, 9 out of 10 being renal glomeruli, and 1 out of 10 being large lectin-labelled non-glomerular structures that could be excluded by the Vobj/Vell ratio segmentation. Taken together, these successive segmentations ensured the automated identification of the renal glomeruli that excluded most of the false positive identification with a small proportion of false negative ones.

### 4.3. Number, Volume and Morphology of the Glomeruli

Our results showed a number of renal glomeruli of 275 per mm^3^ of cortical renal tissue, representing nearly 2% of the renal volume. To our knowledge, no previous study has been published on the relative glomerular volume in mice. Regarding the numerical density, considering an average calculated kidney volume of 150 mm^3^, with 75% being the cortex, the estimated average number of cortical glomeruli per kidney is around 30,000. Previous studies in mice have estimated an average number of glomeruli per kidney to be around 12,000 [11,18]. This difference may be explained by the techniques used in these studies to count the glomeruli, based on the purification of renal glomeruli and manual counting, or MRI counting.

The mean surface area was equal to nearly 21,500 µm², and the mean volume around 63,000 µm^3^. To our knowledge, there is no previous published 3D data on the mean glomerular surface area. However, previous studies have estimated, in mice, the mean glomerular volume with a wide range going from 80,000 µm^3^, a value close to our own ones, to 285,000 µm^3^ [11,19]. This large difference may be due to the imaging technique resolution used, as the lowest values were obtained by light-sheet or confocal microscopy, high resolutive techniques, and the higher ones were obtained by MRI, a low resolutive technique. To our knowledge, the analysis of the frequency distribution of individual surface area and volume of renal glomeruli has not been done in previous studies. The interest of such an analysis is that it provides a more precise description of the population of glomeruli than the mean value, and it would allow the identification of possible alteration of its characteristics (e.g., greater heterogeneity) in pathological conditions.

The compactness, or normalized isoperimetric ratio, is a quantitative morphological parameter useful to characterize the shape of globular structures, such as the renal glomeruli. However, to the best of our knowledge, it has not been used in previous studies to describe the shape of the glomeruli. Our results showed that the glomerular compactness, close to 0.1, is far from 1, the value for a perfect sphere. This is supported by the glomerulus surface reconstruction shown in Figure 3, that showed that the glomerulus surface is complex. Renal glomeruli are far from spheric objects, and this should be taken into account in morphological studies.

## 5. Conclusions

In conclusion, our study proposes a standardized methodology for 3D imaging of mouse kidneys, the automated identification of the renal glomeruli and the visually-independent delimitation of their external surface that allows their 3D morphological quantitative characterization. It might be useful for the identification of morphological alterations of the glomeruli in animal models of renal diseases in comparison with healthy conditions. Being based on an open source software, the whole process is easily accessible for biologists. Additionally, the application of this method to C57BL/6J mice, a strand largely used in biomedical studies, provides a set of qualitative and quantitative data on glomeruli in normal mouse kidneys.

## Figures and Tables

**Figure 1 sensors-21-07440-f001:**
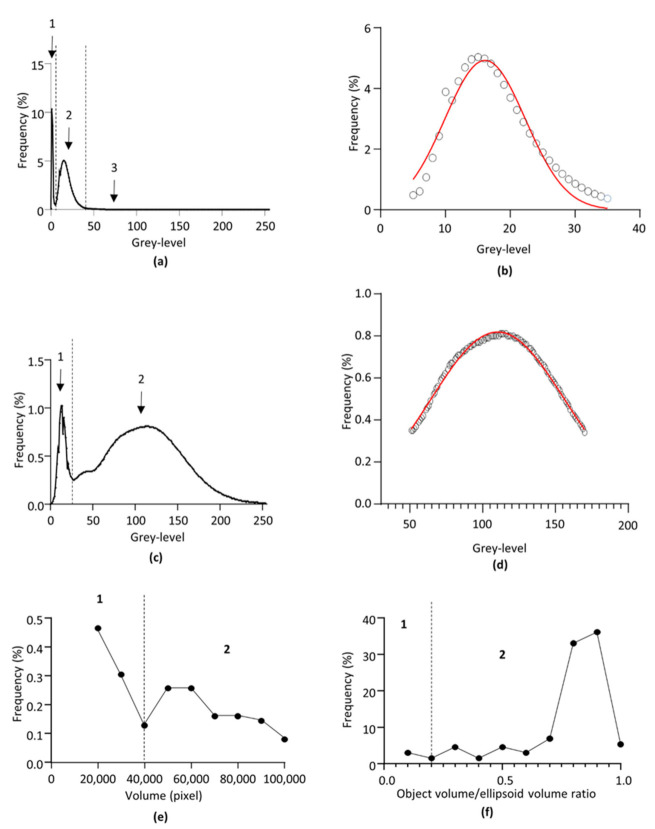
Thresholding process. (**a**) Representative frequency distribution curve of the voxel greyscale value from one lectin-labelled kidney after 3D imaging and 8-bit format conversion. Greyscale values range from the darkest (0) to the brightest (255) voxels of the image. 1: background. 2: non-labelled renal structures. 3: lectin-labelled structure. (**b**) Gaussian fit of the subset of pixels corresponding to the renal tissue from curve (**a**,**c**). Representative frequency distribution curve of the voxel greyscale value from one non-lectin-labelled kidney after 3D imaging and 8-bit format conversion. Greyscale values range from the darkest (0) to the brightest (255) voxels of the image. 1: background. 2: non-labelled renal structures. (**d**) Gaussian fit of the subset of pixels corresponding to the renal tissue from curve (**c**,**e**). Representative frequency distribution curve of the selected objects after greyscale segmentation. Black dots are the center of each frequency class. Vertical dotted line indicates the inflexion point chosen as volume threshold. 1: excluded small lectin-labelled objects. 2: selected big lectin-labelled objects. (**f**) Representative frequency distribution curve of the ratio between the real objects volume and the theoretical ellipsoid volume of the objects selected after volume-based segmentation. Black dots are the center of each frequency class. Vertical dotted line indicates the inflexion point chosen as the shape-based threshold. 1: excluded non-ellipsoid shaped objects. 2: selected ellipsoid shaped objects.

**Figure 2 sensors-21-07440-f002:**
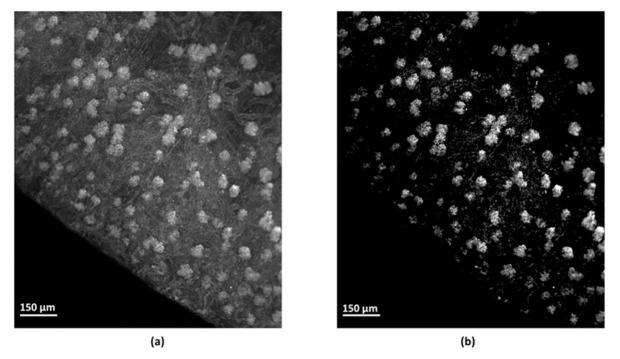
Segmented images of the renal tissue and glomeruli. (**a**) Representative renal cortex image after greyscale-based segmentation of the non-renal background. (**b**) Representative grey-level segmented image of the lectin-labelled structures after greyscale-based segmentation of the non-labelled renal tissue. (**c**) Representative volume segmented image of large lectin-labelled structures after the volume-based segmentation. (**d**) Representative shape segmented image of ellipsoid-shaped objects, i.e., renal glomeruli, after shaped-based segmentation. Lower left scale bar measures 150 µm.

**Figure 3 sensors-21-07440-f003:**
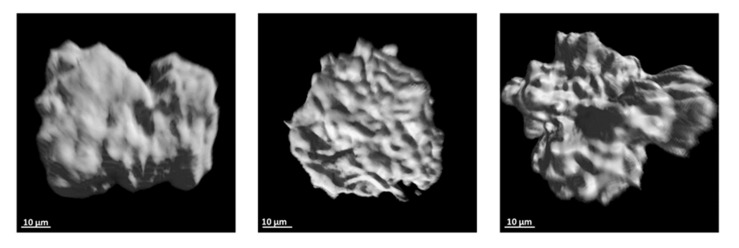
Reconstructed surface of renal glomeruli. Representative images of the surface glomeruli. Each glomerulus was obtained from one of the six left kidneys studied.

**Figure 4 sensors-21-07440-f004:**
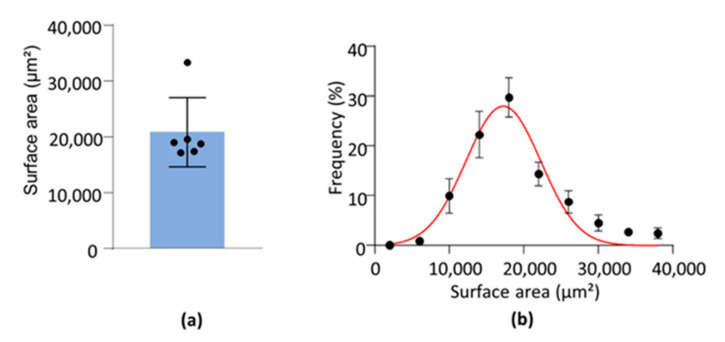
Glomerular surface area, volume and compactness (**a**,**c**,**e**). Mean glomerular surface area in µm² (**a**), mean glomerular volume in µm^3^ (**c**), and mean glomerular compactness (**e**). Each dot corresponds to one sample value (*n* = 6). Blue columns are mean values. Error bars represent SD (**b**,**d**,**f**). Relative frequency distribution of individual glomerular surface area (**b**), volume (**d**), and compactness (**f**). Dots and error bars represent the mean and SEM. Red curves are Gaussian fits of the parameter distribution (see Table 1).

**Table 1 sensors-21-07440-t001:** Surface area, volume and compactness. Values were obtained by a Gaussian non-linear regression of frequency distribution of data from six mouse left kidneys (see Figure 4b,d,f). Estimated mean (*μ*) and standard deviation (*σ*) of the Gaussian distribution (see Equation (5)) of each parameter are expressed as mean ± SEM.

Parameters	Values
Surface	
R²	0.69
*μ*	17,249 ± 870 µm²
*σ*	5014 ± 930 µm²
Volume	
R²	0.51
*μ*	53,579 ± 4200 µm^3^
*σ*	22,057 ± 4500 µm^3^
Compactness	
R²	0.41
*μ*	0.055 ± 0.01
*σ*	0.036 ± 0.01

## Data Availability

The data presented in this study are available on request from the corresponding author.

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
