# Peer review of "Computational Identification and 3D Morphological Characterization of Renal Glomeruli in Optically Cleared Murine Kidneys"

_sensors, 2021, doi:10.3390/s21227440_

Round 1
Reviewer 1 Report
The authors attempted to establish an accessible methodology for the objective identification and 3D morphological characterization of renal glomeruli in mice. The method successfully allows the identification of the renal glomeruli and their 3D morphological characterization.
Although the method is consisted of conventional software, the result would be useful for the researchers.
Reviewer 2 Report
The aim of this manuscript is to examine an accessible methodology for 3D morphological characterization of renal glomeruli in mice. 3D imaging of the renal cortex was performed by light-sheet microscopy on iDISCO+ optical cleared kidneys of 6 C57BL/6J mice after labeling of the capillary endothelium by lectin injection. 3D images were processed with the open-source software ImageJ, and statistical analysis was done with GraphPad Prism. The authors provide the average data were expressed as mean ± SD. The number of glomeruli was equal to 283 ± 35 per mm3 of renal tissue, representing 1.78 ± 0.49% of the tissue volume. The surface area, volume, and compactness were respectively equal to 20830 ± 6200 µm², 62280 ± 14000 µm3, and 0.068 ± 0.026. The proposed standardized methodology allows the identification of the renal glomeruli and their 3D morphological characterization and is easily accessible for biologists.
I congratulate the authors for such an interesting manuscript, well-structured and trendy. I really appreciate the supplementary materail on image processing commands and functions
Reviewer 3 Report
Research on the computational identification and 3D morphological characterization of renal glomeruli in optically cleared murine kidneys is valuable. Authors presents and proves an accessible methodology for the objective identification and 3D morphological characterization of renal glomeruli in mice. The result achieved by the authors is standardized methodology for the identification of the renal glomeruli and their 3D morphological characterization is novel. It is important to note that methodology is based on the open source software and is easily accessible for biologists worldwide.
The proposed methodology is thoroughly described, image processing commands, functions ant application steps are described in supplementary document – this is a strong part of the study.
Few comments to consider that could improve the publication:
- Figure 1 shows thresholding process and graphs (c) and (d) are showing the evidence by control experiments (non-lectin-labelled kidneys) that intensity distribution of the non-labelled tissue follows Gaussian normal distribution. However, it seems that graph (c) is showing normalized distribution (there is no mention about this in the text) as grey-levels of background and non-labelled renal structures in (a) and (c) differs approximately by ratio of 5. This difference is distracting and requires clarification.
- In (e) and (f) graphs of Figure 1 representative frequency distribution curves after segmentation are presented. Why they are discrete? Showing smooth curves would be much more prove for the thresholds selections – especially in (f) graph.
